# The Association between Salt Taste Perception, Mediterranean Diet and Metabolic Syndrome: A Cross-Sectional Study

**DOI:** 10.3390/nu12041164

**Published:** 2020-04-22

**Authors:** Nikolina Nika Veček, Lana Mucalo, Ružica Dragun, Tanja Miličević, Ajka Pribisalić, Inga Patarčić, Caroline Hayward, Ozren Polašek, Ivana Kolčić

**Affiliations:** 1University of Split School of Medicine, Šoltanska 2, 21 000 Split, Croatia; veceknika@gmail.com (N.N.V.); ruzica.dragun@yahoo.com (R.D.); ajka.relja@mefst.hr (A.P.); 2Department of Pediatrics, Medical College of Wisconsin, Milwaukee, WI 53226, USA; lmucalo@mcw.edu; 3Department of Endocrinology and Diabetology, University Hospital Center Split, Šoltanska 1, 21 000 Split, Croatia; tanja.milicevic2@gmail.com; 4Bioinformatics and Omics Data Science Platform, Berlin Institute for Medical Systems Biology, Max-Delbrück Center for Molecular Medicine, 13092 Berlin, Germany; inga.patarcic@mdc-berlin.de; 5MRC Human Genetics Unit, IGMM, University of Edinburgh, Edinburgh EH4 2XU, UK; caroline.hayward@igmm.ed.ac.uk; 6Department of Public Health, University of Split School of Medicine, Šoltanska 2, 21 000 Split, Croatia; opolasek@mefst.hr

**Keywords:** salt taste perception, taste threshold, sodium chloride, metabolic syndrome, Mediterranean diet

## Abstract

Metabolic syndrome (MetS) is a widespread disorder and an important public health challenge. The purpose of this study was to identify the association between salt taste perception, Mediterranean diet and MetS. This cross-sectional study included 2798 subjects from the general population of Dalmatia, Croatia. MetS was determined using the Joint Interim Statement definition, and Mediterranean diet compliance was estimated using Mediterranean Diet Serving Score. Salt taste perception was assessed by threshold and suprathreshold testing (intensity and hedonic perception). Logistic regression was used in the analysis, adjusting for important confounding factors. As many as 44% of subjects had MetS, with elevated waist circumference as the most common component (77%). Higher salt taste sensitivity (lower threshold) was associated with several positive outcomes: lower odds of MetS (OR = 0.69; 95% CI 0.52–0.92), lower odds for elevated waist circumference (0.47; 0.27–0.82), elevated fasting glucose or diabetes (0.65; 0.45–0.94), and reduced HDL cholesterol (0.59; 0.42–0.84), compared to the higher threshold group. Subjects with lower salt taste threshold were more likely to consume more fruit, and less likely to adhere to olive oil and white meat guidelines, but without a difference in the overall Mediterranean diet compliance. Salt taste intensity perception was not associated with any of the investigated outcomes, while salty solution liking was associated with MetS (OR = 1.85, CI 95% 1.02–3.35). This study identified an association between salt taste perception and MetS and gave a new insight into taste perception, nutrition, and possible health outcomes.

## 1. Introduction

Metabolic syndrome (MetS) is a cluster of synergistic risk factors, such as abdominal obesity, arterial hypertension, hyperglycemia, and dyslipidemia that contribute to cardiovascular disease (CVD) and mortality. There is a surge in the prevalence of all of the components of MetS, causing a worldwide pandemic and implicating both clinical and public health [1]. Given that there are still several definitions in use, which differ in their cut-off values for MetS components, the prevalence of MetS in the literature ranges anywhere between 10% and 84%, depending both on the characteristics of the sample and definition used [2]. A majority of the studies indicate that 15% to 40% of the adult population in most countries can be characterized as having MetS [3,4,5,6,7]. Mediterranean countries also exhibit high MetS prevalence, ranging from one quarter to one third of the population [8,9,10]. Unfortunately, Croatia is not at all an exception to this epidemiological situation. Previous studies have shown rather high burden of MetS in Croatian population, with crude prevalence ranging from 36% to as high as 60% in the Mediterranean region of the country [11], and 40% in the continental Croatia [12]. 

The main driving force contributing to such high prevalence of MetS is increase in obesity due to overconsumption of calorie-dense foods and drinks, with simultaneous decrease in physical activity levels and an alarming proportion of sedentary lifestyle [13]. The unprecedented increase in obesity worldwide has resulted from the perfect storm conditions enabled by industrial production of highly processed food, drift from the traditional food consumption practices, overall labor-saving technological advances, environmental, socio-economic and demographic changes. According to the global survey on obesity in 195 countries, 604 million adults and 108 million children were obese in 2015 [14]. Since 1980, prevalence of obesity doubled in 73 countries and increased in most other countries [14]. The Mediterranean region displays a particularly worrisome trend in childhood obesity. A recent study showed an increase in the prevalence of overweight and obesity from 22.9% in 1999 to 25.0% in 2016 among children aged 2 to 13 years in the Mediterranean part of Europe [15]. One of the explanations for this trend is departure from the traditional lifestyle and Mediterranean diet, especially in younger people from Mediterranean countries [16,17]. Interestingly, it was found that change in the food supply in the Mediterranean area, especially more readily available mass-produced food from the long supply chain (opposite from the local food markets) was associated with MetS [18]. These trends are very misfortunate and represent a double missed opportunity, because Mediterranean diet was shown to have the capacity for preventing the development of metabolic syndrome, as well as the ability to reverse it in people with or without type 2 diabetes [19,20]. 

Along with economic, social and environmental factors, taste perception is a major determinant of dietary choices and its impact on obesity has been previously studied. However, study results on this topic are still contradictory and inconclusive. Obese adults were reported to consume more salty foods and to have reduced salt sensitivity and higher salt preference [21,22,23]. Additionally, obese women showed decrease in both olfactory and taste capacity, including salt taste, compared to normal weight women [24]. On the other hand, adolescents with early onset and severe obesity displayed lower recognition thresholds, indicating higher acuity, and higher sensitivity for both sucrose and salt compared to the non-obese adolescents [25]. Finally, some studies found no association between body composition and salt taste sensitivity threshold [26], and no association between obesity and salt liking [27]. On the contrary, fat-liking was found to be associated with an increased risk of obesity [27]. Moreover, animal models have shown that high-fat diet resulted in obesity and pronounced loss of taste buds, indicating that taste loss could be a metabolic consequence of the obese state [28]. 

The exact mechanism on how greater sodium consumption could contribute to higher body weight remains unclear. Some authors propose that sodium intake is often accompanied by higher consumption of energy-dense foods and soft drinks [29]. The well-recognized link between high dietary salt intake, arterial hypertension and endothelial dysfunction brought the salt taste preference into focus [30]. However, studies investigating the association between salt taste and MetS are very scarce in the literature [31,32]. Furthermore, an even greater paucity exists at the intersection of salt taste sensitivity, nutrition and MetS research. Such studies would provide valuable information about the factors contributing to the MetS, which is very important for targeted prevention and treatment approaches. Prevention of MetS should be strongly emphasized since numerous studies have demonstrated that people with MetS have a 5-fold increase in risk of type 2 diabetes and a 3-fold increase in risk of CVD and related mortality, as well as increased risk for cancer [2,33]. Given these serious health consequences of MetS, any additional insight that illuminates its contributing factors is advantageous and welcome. 

The aim of this study was to examine the association between salt taste threshold and suprathreshold perception and MetS components in the general population of Dalmatia, Croatia. Additionally, we examined the adherence to the Mediterranean diet according to the salt taste perception.

## 2. Materials and Methods

### 2.1. Study Participants

This cross-sectional study included subjects from the “10,001 Dalmatians” project [34], which was previously described in details [11]. For the purposes of this study, a sub-sample of 2798 subjects was used from three Dalmatian settlements: the Island of Vis (*n* = 390, sampled in 2011), the Island of Korcula (*n* = 1908, sampled during 2012–2016 period) and the City of Split (*n* = 500; sampled in 2012–2013). A population-based sample was gathered via direct postal invitations, radio announcements and support from the local general practitioners and local government officials. The only exclusion criterion was age of <18 years old. All potential subjects were informed on the study aims and goals, expected benefits and risks, after which those who decided to participate have signed the informed consent. The study was conducted in accordance with the Declaration of Helsinki, and approved by the Ethical Board of the University of Split School of Medicine (2181-198-03-04/10-11-0008).

### 2.2. Data Collection and Measurements

Each subject provided a fasting blood sample, filled a self-administered questionnaire and had blood pressure and anthropometric measurements done. Trained medical professionals (medical doctor or research nurse) collected data on medical history and previous diagnoses, as well as on medications being used for: hypertension, coronary heart disease (CHD), cerebrovascular insult (CVI), type 2 diabetes, hyperlipidemia, cancer, bipolar disorder and gout.

The questionnaire included questions on demographic characteristics (age and gender), socioeconomic status (education and material status), smoking, alcohol consumption, physical activity, and dietary habits. Education was classified in three categories based on the number of years of completed schooling (primary, secondary, and university level). Material status was based on the subjects’ material possessions and classified into quartiles, as described previously [11]. Namely, subjects answered to 16 questions about their material possessions (heating system, wooden floors, video/DVD recorder, telephone, computer, two TVs, freezer, dishwasher, water supply system, flushing toilet, bathroom, library with more than 100 books, paintings or other art objects, a car, vacation house or second apartment, boat). Based on the sum of all positive responses, subjects were divided into quartiles.

Based on smoking habits, we divided subjects in active smokers (for whom we calculated the number of pack-years, by multiplying the number of cigarettes smoked per day with the number of years they smoked), ex-smokers (stopped smoking more than 1 year ago) and non-smokers. Alcohol consumption was measured as the total number of units of alcohol ingested per week, while physical activity was self-assessed as light, moderate or intensive. 

The Mediterranean Diet Serving Score (MDSS) was calculated as suggested by Monteagudo et al., based on the food frequency questionnaire consisting of 55 questions [35]. Shortly, this scoring approach requires daily consumption of cereals, vegetables, fruit and olive oil (in each main meal), one or two daily servings of nuts and dairy products, daily moderate alcohol intake (ideally a glass of wine per day), consumption of fish and legumes several times per week, while other meats and sweets should be consumed rarely, once or twice per week [35]. The maximum MDSS value is 24 points, and the cut-off value of ≥14 points was used to define compliance with the principles of the Mediterranean diet [11]. Additionally, we asked subjects about their habits of adding salt before tasting food, and they could have responded as never, occasionally, often or almost always.

Blood pressure was measured in a sitting position after a rest period of at least ten minutes. We measured the blood pressure twice in each subject in order to calculate the average value, which was used in the analysis.

Anthropometric measures included body height and mass, measured using a combined stadiometer and a scale instrument (model 704, Seca GmBH & Co., Hamburg, Germany), while waist and hip circumferences were measured in millimeters using an inelastic measuring tape. Using these measures we have calculated body mass index (BMI), waist-to-hip ratio (WHR) and waist-to-height ratio (WHtR), as relative estimates of central obesity. During the anthropometric measurement, subjects were dressed in underwear or light clothing.

### 2.3. Biochemistry Measurements and Metabolic Syndrome Definition

After blood collection, the sample was processed in a field laboratory and stored in a −80 °C freezer. Biochemical analysis was performed at accredited Brayer Polyclinic laboratory in Zagreb using standard methods for determining biochemical parameters. In this study, we used data on fasting glucose (mmol/L), triglycerides (mmol/L), total cholesterol (mmol/L), LDL cholesterol (mmol/L), HDL cholesterol (mmol/L) and HbA1c (%).

Metabolic syndrome was defined according to the Joint Interim Statement definition [36]. The subject had to have at least three of the following criteria: elevated waist circumference (≥80 cm for women and ≥94 cm for men), elevated triglycerides (≥1.7 mmol/L) or using medications, reduced HDL concentration (<1.0 mmol/L in men, <1.3 mmol/L in women) or using medications, elevated systolic and/or diastolic blood pressure (≥130/85 mmHg) or using medications for hypertension, elevated fasting glucose (≥5.6 mmol/L) or using medications for diabetes [36].

### 2.4. Measurement of Salt Taste Perception

Salt taste perception was assessed by threshold and suprathreshold testing designed according to the ISO standards [37], and performed by trained researchers. Subjects had to restrain from chewing gum, smoking, eating and drinking anything except water, at least half an hour before testing. All of the tests were performed using water solutions of table salt. Solutions were prepared daily and kept at room temperature. Due to the conditions of the field testing away from the laboratory, we used a standardized commercial mineral water, with following content: 64.2 mg/L of Ca^2+^, 32.1 mg/L of Mg^2+^, 1.7 mg/L of Na^+^, and 2.8 mg/L of Cl^−^.

For salt taste recognition threshold, we used five concentrations, starting with the weakest solution and equally increasing concentrations (0.22 log increment). These solutions, each in an individual volume of 10 mL, were presented to the subjects starting from the lowest concentration of 8.21 mmol/L (0.48 g of NaCl dissolved in 1 L of water), followed by 13.69 mmol/L, 22.81 mmol/L, 38.02 mmol/L, while the highest concentration was 63.37 mmol/L. We performed a pilot testing (*n* = 32) using these concentrations in order to confirm that they are appropriate for use in the general population across different ages, identifying both people who recognize the lowest concentration and those not recognizing the highest concentration. 

Subjects were blinded to the taste quality presented to them and increasing concentrations of solutions were used until the point when they correctly recognized salty taste. Subjects were instructed to taste the solution for a couple of seconds and they were allowed to swallow the solution before providing their answer. The correct answer was denoted as the ordinal number of the solution, starting with the number 1, which marked the lowest concentration and number 5 marked the highest concentration, while number 6 was used in cases when subjects didn’t recognize salty taste of the solution with the highest concentration. Between testing solution presentations, subjects were instructed to rinse their mouth with the same water used for preparation of testing solutions. Subjects were presented with only one solution at a time in order to make the testing procedure overall less time consuming and less cumbersome, especially for elderly subjects. With each solution presentation, subjects were asked whether the solution tasted like plane water or something else, and in case of confirmatory answer, they were asked to identify the quality of the taste (sweet, salty, sour or bitter). Because of this simplified testing procedure, unlike the usual 3-alternative forced choice or 2-alternative forced choice, each subject performed two recognition threshold testing rounds, with a break of at least of 30 min in between them. Based on these two testing responses, a geometric mean was calculated using the ordinal numbers of correctly recognized solutions. Those subjects who had a geometric mean result of ≤2.0 were considered as having a lower salt taste recognition threshold, which corresponds to the higher sensitivity and acuity. A geometric mean between 2.1 and 4.0 was considered as intermediate threshold, while those subjects with a result of ≥4.1 and those who didn’t recognize the highest salt concentration at both testing times were considered as having a higher taste threshold and lower salt taste acuity. 

After threshold testing, suprathreshold salt taste perception was tested as perceived intensity and hedonic response (liking), using a 10 mL of table salt solution with the concentration of 137 mmol/L (8 grams of NaCl per liter). This concentration is slightly more than double the highest threshold solution concentration (3.7 g/L), and it was used before as the highest concentration for threshold testing [38], pointing to be a possible concentration for suprathreshold testing, but not too concentrated to be off-putting. Suprathreshold measurements were available in a subsample of the subjects consisting of 1155 people sampled after 2012 (926 subjects from Korcula and 229 subjects from Split subsample). The Labeled Magnitude Scale (LMS) was used to estimate the taste intensity perception [39]. In short, we placed words describing the intensity of the salty taste along a vertical line, without any numeric markers. The words “no sensation” were placed at the start of the line (0 mm), “barely detectable” at 2 mm, “weak sensation” at 7 mm, “moderate” at 20 mm, “strong” at 40 mm, “very strong” at 61 mm, and “strongest imaginable” was placed at 114 mm from the beginning of the line [37]. Subjects have practiced using the scale in at least one tasting attempt, after which they placed their final mark on the LMS line immediately after tasting the solution. Their intensity response was measured in millimeters as the distance of the subjects’ mark relative to the beginning of the scale. Based on the corresponding wording along the line, we have divided subjects into three groups in order to simplify the analysis. Subjects who marked their response between 0 and 39 mm on the LMS line were considered as having felt nothing to medium strong intensity (lower perceived intensity). Subjects responding between 40 to 61 mm found the solution to be strong to very strong, and those who responded between 62 and 114 mm thought that the solution was extremely strong in intensity.

Hedonic perception (liking) was tested using the same suprathreshold salt solution concentration, using the Labeled Affective Magnitude scale (LAM) [40]. LAM scale is also a vertical line with a total length of 100 mm, where semantic labels “greatest imaginable dislike”, “neither like nor dislike”, and “greatest imaginable like” were placed at 0, 50 and 100 mm, respectively, but without displayed numbers [40]. Subjects were asked to make a mark indicating how much they liked or disliked the taste of the concentrated salt solution. Their responses were turned into a numeric variables in the way that the middle of the scale was regarded as a 0 mm, and negative responses were below that point (the start of the line was marked as −50 mm and it indicated “greatest imaginable dislike”), while positive response were above the middle point (up to +50 mm, indicating “greatest imaginable like”). Based on these distances, we have divided subjects into three groups; those who disliked the solution (response between −50 to −10 mm), those who neither liked nor disliked it (−9 to +9 mm), and those who liked concentrated salty solution (+10 to +50 mm).

### 2.5. Statistical Analysis

Categorical variables were presented as absolute numbers and percentages. Numerical variables were tested for normality using Kolmogorov–Smirnov test and central tendency was presented using medians and interquartile ranges (IQR), due to mostly non-normal data distribution. Differences between groups were tested using chi-square test for categorical variables, and Mann–Whitney U test or Kruskal–Wallis tests were used for numerical variables, depending on the number of groups. Spearman correlation test was used to identify bivariate correlation between age, threshold and suprathreshold salt taste perception, and the frequency of adding salt before tasting the food.

The association between the presence of the MetS and salt taste threshold (full sample, *n* = 2798), salt taste intensity and hedonic perception (subsample, *n* = 1155) was tested by multivariate logistic regression analysis. Several logistic regression models were created and adjusted for confounders. Prevalent MetS and each of the five MetS components were considered to be dependent variables. The association between independent variables (taste threshold, taste intensity and taste hedonic perception) and dependent variables was adjusted for age, gender, place of residence (Island Vis, Island Korcula, City of Split), education level (primary, secondary, university), quartiles of material status, BMI (in all regression models except for elevated waist circumference), smoking (never-smokers, ex-smokers, active smokers), alcohol intake (units/week), physical activity (low, moderate, intensive), Mediterranean Diet Serving Score (MDSS), and adding salt before tasting food (never, occasionally, often, almost always).

Additionally, the association between the Mediterranean diet (MDSS ≥14 points) and it’s components (dependent variables) and salt taste perception (three independent variables) was tested using multivariate logistic regression analysis. Models were adjusted to the same confounding variables, but excluding the MDSS score. 

Significance level was set at *p* < 0.05. Data analysis was conducted using IBM SPSS Statistics for Windows, v21.0 (IBM, Armonk, NY, USA).

## 3. Results

This cross-sectional study included 2798 subjects from the Island of Vis, the Island of Korcula and the City of Split. According to their salt taste recognition threshold, we divided subjects into three categories, where lower taste threshold indicated higher taste acuity. Subjects with higher salt taste threshold were on average older, had higher proportion of men, lower education level, higher anthropometric indices, and highest average values for all of the MetS constituent components, except for HDL cholesterol (Table 1). There were no differences in habits, except in the Mediterranean diet adherence and adding salt before tasting food. Subjects with higher salt threshold added salt to their food more frequently compared to subjects with both lower and intermediate threshold (Table 1).

Subjects with MetS were on average older, had higher proportion of men and Vis Island inhabitants, lower education level, higher anthropometric indices, and higher average values for all of the MetS constituent components, as well as other biochemical parameters (Appendix A, Table A1). Subjects with MetS were less frequently active smokers, but with higher average pack-years among smokers, had slightly higher proportion of subjects with intensive level of physical activity, greater average score of the Mediterranean diet adherence, and they added salt to their food more frequently compared to subjects without MetS (Appendix A, Table A1).

Salt taste threshold was correlated negatively with intensity perception and positively with age and the habit of adding salt before food tasting (all *p* < 0.001). Additionally, salt taste intensity perception was correlated negatively with age and hedonic perception (both *p* < 0.001) (Table 2). 

We observed high prevalence of MetS, with differences according to the salt taste threshold sensitivity, where 57.9% of subjects with lower taste acuity had MetS, compared to 38.3% of subjects with higher taste acuity (lower salt taste threshold) (Table 3). Elevated waist circumference was the most common metabolic syndrome component in all threshold sensitivity groups (as high as 79.5%), followed by elevated blood pressure (up to 56.2% in subjects with higher threshold) (Table 3). The only MetS component without significant difference between threshold sensitivity groups was HDL cholesterol.

After stratification according to the age, only middle-aged subjects (35–65 years old) presented with higher salt taste threshold more frequently in subjects with MetS (18.6% vs. 12.0%), and less frequently with lower threshold compared to the subjects without MetS (38.6% vs. 43.2%). Similar results were present in the subgroup of subjects older than 65 years, but with borderline insignificant result (*p* = 0.056) (Appendix A, Table A2). There were no differences in either salt taste intensity or hedonic perception between subjects with MetS and those without it, in any of the age groups (Appendix A, Table A2).

There was a borderline insignificant result in the Mediterranean diet compliance between subjects with different taste threshold perception (Appendix A, Table A3). We also observed that subjects with higher salt taste thresholds more frequently complied with the Mediterranean pyramid recommendations for olive oil, legumes, fish, and white meat, while they less frequently complied with fruit and potatoes guidelines, compared to subjects in lower taste threshold group (Appendix A, Table A3). Similar results were obtained in the regression analysis adjusted for confounding factors, where subjects with lower salt taste threshold were more likely to consume fruit several times a day (OR = 1.52, 95% CI 1.16–1.97; *p* = 0.002), the same as subjects with intermediate threshold (OR = 1.41, 95% CI 1.09–1.81; *p* = 0.008), compared to subjects with higher threshold (Table 4). The opposite was recorded for olive oil and white meat consumption, while the result for fish was borderline insignificant (OR = 0.76, 95% CI 0.58–1.00; *p* = 0.053). There were no differences in consumption of vegetables, legumes, red meat and sweets, or in overall compliance to the Mediterranean diet between subjects with lower and higher salt taste thresholds (Table 4). There were also no differences in the Mediterranean diet or in food groups compliance with regard to the salt taste intensity nor with hedonic perception (Table 4).

Subjects with lower salt taste threshold had lower odds of having elevated waist circumference (OR = 0.47, 95% CI 0.27–0.82; *p* = 0.008; fully adjusted model), the same as for having elevated fasting glucose or diabetes (OR = 0.65, 95% CI 0.45–0.94; *p* = 0.022), compared to higher threshold group (Table 5). Subjects with both lower and intermediate threshold had lower odds of having reduced HDL cholesterol (OR = 0.59, 95% CI 0.42–0.84; *p* = 0.003 and OR = 0.65, CI 95% 0.47–0.91; *p* = 0.011, respectively) and lower odds of having MetS (OR = 0.69, 95% CI 0.52–0.92; *p* = 0.013 and OR = 0.75, 95% CI 0.57–0.99; *p* = 0.044, respectively) (Table 5). 

Salt taste intensity perception did not show significant association with metabolic syndrome and its components. Only subjects who liked salty solution had higher odds of having metabolic syndrome (OR 1.85, CI 95% 1.02–3.35; *p* = 0.042), compared to subjects who disliked the solution (Table 5). Those subjects who liked salty solution also had a borderline insignificantly higher odds for having elevated blood pressure or diagnosis of hypertension (OR = 1.79, 95% CI 0.97–3.31; *p* = 0.063) and borderline insignificantly lower odds for having reduced HDL cholesterol (OR = 0.48, 95% CI 0.22–1.03; *p* = 0.058) (Table 5).

Pictorial presentation of the main findings of the study, with the results from adjusted logistic regression models are presented in Figure 1.

## 4. Discussion

The most important and new findings of this study include identification of the lower odds for MetS and most of the MetS components in subjects with higher salt taste sensitivity (lower threshold), as well as the association of salt taste threshold with several Mediterranean diet food groups in the general population. Contrary to this, we found no indication of the association between suprathreshold salt taste perception and these outcomes, except for higher odds of MetS in subjects with higher liking of salty solution.

MetS is a common disorder in the general population [41]. The same situation is present in Croatia. As many as 44% of subjects included in this study had MetS, which is similar to previously reported MetS prevalence in both adult population in Croatia [42,43] and in obese children and adolescents [44], while some studies found crude MetS prevalence to be even greater than 55% [45]. Some of the differences between these studies can be explained by different diagnostic criteria being used and different population groups included. The MetS is associated with many adverse outcomes, such as increased risk of cardiovascular disease, diabetes, chronic kidney disease and total mortality [46,47]. This makes MetS a very important public health challenge and a research target, in order to identify risk factors behind its development and useful approaches in prevention and treatment. Many of the MetS risk factors have been identified and repeatedly confirmed, such as poor nutrition and lack of physical activity. Determinants influencing these risk factors are now becoming increasingly important, and taste perception is surely among them [32,48]. Taste and olfaction form the basis of flavor perception, and as such, they are well-recognized and major predictors of food choices, dietary patterns, body composition and consequent health outcomes [49]. The sense of taste has been extensively studied and many determinants of individual differences in taste perception have been identified, such as genetic factors, age, habits and lifestyle factors, alongside with various pathologies and metabolic diseases, such as obesity [50]. Salt taste perception was investigated to a lesser extent, and most commonly in relation to the salt sensitivity (change in blood pressure depending on the change in salt intake), hypertension and salt intake [51,52]. The association between salt intake and hypertension was indeed extensively studied [30]. Salt taste perception was substantially less frequently investigated, especially in association with the MetS. There are only a handful of studies published so far on this topic [31]. Hence, we aimed to fill this gap and examine the association of both salt taste threshold sensitivity and suprathreshold perception with different health outcomes included in the MetS definition. Namely, based on the regression analysis, we found that subjects with lower salt taste threshold, indicating higher salt taste sensitivity, had 31% lower odds of having MetS (OR = 0.69; 95% CI 0.52–0.92). They also had 53% lower odds for elevated waist circumference (OR = 0.47; 95% CI 0.27–0.82), 35% lower odds for having elevated fasting glucose or diabetes diagnosis (OR = 0.65; 95% CI 0.45–0.94), and 41% lower odds for having reduced HDL cholesterol (OR = 0.59; 95% CI 0.42–0.84), compared to the higher threshold group, while the result for elevated triglycerides was borderline insignificant (OR = 0.76; 95% CI 0.57–1.02). These results confirm previous results of increased salt taste threshold in subjects with MetS compared to the subjects without MetS, which was independent of sex, age and BMI [31]. Another study found a positive association between sodium excretion, indicating higher intake, and components of MetS, such as blood pressure, waist circumference, triglycerides, and fasting glucose and an inverse association with HDL cholesterol [53]. Additionally, subjects with higher sodium excretion also had a higher body fat percentage, body fat mass, and insulin levels, pointing to the high-salt diet as a significant risk factor for MetS [53]. Additionally, several studies have identified the association between salt taste perception and obesity, which is a fundamental MetS component. For instance, one of these studies, including only healthy adults, showed that salt taste threshold was higher in people with higher BMI, with a similar result for olfactory threshold, indicating that increasing BMI was associated with a decrease in olfactory and taste sensitivity [54]. A decreased taste capacity was found with increase in visceral fat, with a negative correlation between salt taste threshold and BMI, total fat mass and visceral fat, as well as with insulin, leptin, glucose, and HDL cholesterol in healthy women [24]. However, there are studies that showed the opposite results. For example, Hardikar et al. found that obese subjects had lower thresholds for sucrose and salt, as well as higher ratings of intensity, indicating a higher sensitivity to sweet and salty tastes, compared to lean subjects [55]. Donaldson et al. pointed that threshold was lower for salt, unchanged for sweet and higher for bitter and sour taste in obese adults [22]. 

We found an association between higher salt taste threshold and elevated blood pressure and/or previous diagnosis of hypertension in bivariate analysis, but this association was not confirmed in multivariate analysis. This was actually the only MetS component not showing the association with salt taste threshold in our subjects, while controlling for important confounding factors. This is in contrast with previous studies, which showed that higher salt taste sensitivity threshold was associated with increased blood pressure [56], even so in women with normal-range blood pressure [57], and also in response to exercise [58]. However, some studies did not manage to demonstrate the association between salt taste threshold and hypertension [59] or between suprathreshold intensity perception and either hypertension or mean blood pressure [60]. An inverse association was reported between salt taste intensity perception and the frequency of adding salt to foods [60]. This habit of adding salt to the food before tasting was rather prevalent in our subjects, and it was positively correlated with both salt taste threshold and hedonic perception, but not with age or with salt taste intensity perception. Such habit should be strongly discouraged because daily salt consumption in food is unequivocally associated with increase in blood pressure and risk for hypertension. Moreover, animal and human studies showed that long-term intake of high-sodium diet increases the risk of obesity, insulin resistance and diabetes development, irrespective of total energy or glucose intake [61,62]. Lanaspa et al. performed an interesting study in which they elaborated potential underlying pathophysiological mechanism by which salt may cause obesity and MetS [63]. They showed that prolonged high-salt diet in mice generated endogenous fructose production by activating hepatic aldose reductase (AR), what resulted in hepatic sorbitol and triglyceride accumulation, as well as serum leptin elevation [63]. Obesity, on the other hand, can be both caused and lead to hedonic eating (eating for pleasure and not for hunger), by disrupting the normal taste input processing [64]. Elevated BMI was found to be related to changes in the brain activity in regions involved in salt taste perception [23]. Indeed, it was shown that salt taste engages various brain regions that modulate reward, taste processing, and executive control in eating [64], possibly resulting in greater salt consumption in overweight/obese people, in association with reduced salt sensitivity and a higher salt preference [23,64]. 

The association between salt taste perception and dietary choices and habits are not well understood or extensively studied. Only a handful of studies have so far examined this topic [65,66]. One such study showed that healthy adults who were hyposensitive to salty taste consumed more bakery and salty baked products, more saturated fat-rich products, and less soft drinks compared to people with higher taste acuity [65]. To our best knowledge, this is the first study to investigate the association between salt taste perception and adherence to the Mediterranean diet. Our results showed that subjects with lower salt taste threshold more frequently complied with the Mediterranean pyramid recommendations for fruit, but less so with the recommendations for consumption of olive oil and white meat (and borderline insignificant for fish), compared to subjects with higher salt taste threshold. After adjusting for important confounding factors, there was no difference in the overall compliance to the Mediterranean diet between lower and higher salt taste threshold group, possibly due to the opposite associations found for fruits and olive oil. 

Overall, compliance with the Mediterranean diet was rather low in our subjects (23%). As we reported previously, it was particularly low in younger age groups, and lower in men compared to women [17]. Unfortunately, this departure from the traditional Mediterranean diet in the population of Dalmatia represents potentially invaluable losses in the domains of population health, environmental sustainability, local economy and cultural heritage preservation [67]. Population health might be on the line already, given that the recent generations of the Adriatic islanders have lost their advantage in life expectancy at birth compared to the mainland population, possibly due to diminishing adherence to the Mediterranean diet and traditional lifestyle [68]. 

Furthermore, the loss of Mediterranean diet represents an immense missed opportunity for primary prevention of CVD. Namely, it was shown that people compliant with the Mediterranean diet had a 30% risk reduction for the major cardiovascular event (myocardial infarction, stroke, or death from cardiovascular causes) [69]. Besides primary and secondary prevention of CVD, Mediterranean diet plays a role in improving health in overweight and obese patients, preventing the increase in weight and waist circumference in non-obese people, and both improving MetS and reducing its incidence [70]. A meta-analysis including 33,847 individuals showed that high adherence to the Mediterranean diet reduced the risk of MetS by 19% [71]. In addition, higher intake of some polyphenols, which have been suggested to be partly responsible for the beneficial effects of the Mediterranean diet, showed an inverse association with blood pressure, fasting plasma glucose, HDL cholesterol and triglycerides [72]. 

Further studies are needed to elucidate the link between salt taste perception and Mediterranean diet adherence, especially since we did not find any apparent association between salt taste intensity or hedonic perception and the Mediterranean diet compliance in our study. The only significant association we have identified between suprathreshold salt taste perception was a 85% increase in odds for MetS presence in subject who positively rated salty solution compared to those who disliked it. Additionally, we identified a suggestive higher odds (borderline insignificant) for elevated blood pressure or hypertension and lower odds for reduced HDL cholesterol in subjects who liked salty solution. Contrary to our expectations and due to the negative correlation between salt taste threshold and intensity perception, we found no significant associations with the perceived intensity rating. For example, a study from Coltell et al. did show an inverse association between higher taste intensity and MetS components [32]. 

As mentioned above, our subjects have reported adding salt to their food before tasting quite frequently, and as many as 32% of subjects said they do it often or almost always. Furthermore, it is known that the average daily salt intake in Croatian population is extremely high, estimated to be as much as 13 g in men and 10 g in women [73]. This could have affected our results substantially. Indeed, previous studies demonstrated plasticity in salt taste perception, pointing to the findings that manipulation with dietary salt intake has the potential to change both salt preference and perception in adults [73]. Once people are habituated to a diet with the certain amount of salt, foods with lower salt content are perceived as less intense. Within the context of a high salt diet, this may lead to the poor acceptance of low salt foods, which may explain in part why adherence to a low sodium diet is initially difficult for most of the people [74]. Additionally, salt taste perception can be influenced by many other characteristics, such as smoking, excessive alcohol intake and age [75].

Our study showed that subjects with higher salt taste threshold were on average older than those with lower threshold. Age was also negatively correlated with salt taste intensity perception. The same was found in a recent study including a large sample from the general population, with even stronger negative association for the higher concentrations of the testants [76]. It is well established that sensory acuity diminishes with age, albeit the sense of smell is more prone to deterioration with age than the sense of taste [77]. Consequently, age-related changes in sensory perception and preference could have a major impact on appetite and food intake. In addition, ageing brings a variety of lifestyle changes as well as a greater number of chronic diseases and associated medications, which can affect taste sensation [78]. 

Several limitations and advantages of this study should be considered. Firstly, this is a cross-sectional study, and therefore causality cannot be assessed in determining taste differences as a cause of MetS. Namely, we cannot elucidate whether subjects with reduced salt taste acuity (increased threshold) developed MetS as a consequence of this sensory characteristic or the MetS was a causal predictor of the salt taste acuity loss. For example, subjects with lower salt taste acuity could also have altered perception of food, which could influence their food choices toward less healthy, more processed salty foods, influencing their body weight and the risk for MetS development. Unfortunately, these habits were not included in the questionnaire, and we can not clarify this further based on the available data. Secondly, data for salt taste intensity and hedonic perception were available only for a subset of subjects, which might have introduced bias of undetermined direction and magnitude. The conditions of the field-testing were not ideal as would have been in the laboratory, which could have resulted in less accurate and precise measures of salt taste perception. Lastly, we used aqueous solution to deliver testant, while most of the salty foods are not consumed in a liquid form, which might have influenced suprathreshold responses and especially hedonic responses of the subjects. For instance, subjects could have rated concentrated salty solution as more intense and less hedonically appealing, compared to the rating they would provide for solid salty foods, such as chips or cured processed meats. This could be behind our result of absent association between suprathreshold salt taste perception and prevalent MetS and its components. The main advantages of this study include a large population-based sample size and the testing of both threshold and suprathreshold salt taste perception. The analysis included numerous confounders related to MetS, such as diet, physical activity, smoking, alcohol consumption and adding salt before tasting food. To the best of our knowledge, this is the first study to examine the association between salt taste threshold sensitivity, intensity perception and hedonic rating, with both MetS and the Mediterranean diet. 

In conclusion, this study adds new insights into the existing body of knowledge about salt taste perception, nutrition and possible health outcomes. Still, there is much to be investigated, given the amount of discrepancies between previous studies, which are limited in number. Given the importance of salt taste in food palatability and associated food choices, as well as its role in propelling overweight and obesity, hypertension, MetS and other health outcomes, further studies are warranted.

## Figures and Tables

**Figure 1 nutrients-12-01164-f001:**
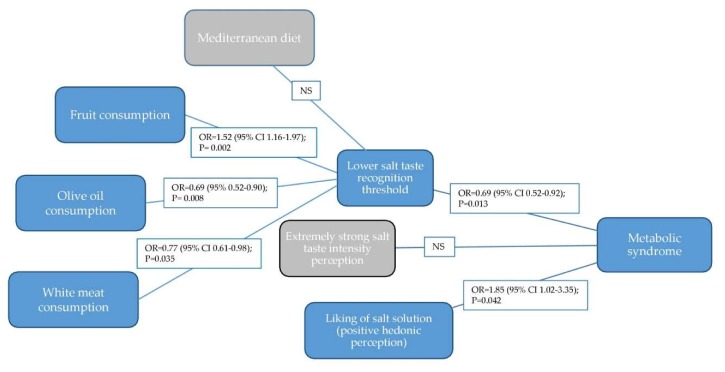
Pictorial presentation of the main findings of the study (results are from adjusted logistic regression model; NS—non-significant).

**Table 1 nutrients-12-01164-t001:** Subjects’ characteristics according to the salt taste recognition threshold perception.

	Lower Salt Taste Threshold/Higher Acuity *n* = 1094	Intermediate Salt Taste Threshold/Acuity *n* = 1236	Higher Salt Taste Threshold/Lower Acuity *n* = 468	*p*
**Socio-demographic characteristics**				
**Age (years); median (IQR)**	52.0 (23.0)	56.0 (21.0)	59.0 (19.0)	<0.001 *
**Gender; *n* (%)**				
**Women**	751 (68.6)	746 (60.4)	260 (55.6)	<0.001 †
**Men**	343 (31.4)	490 (39.6)	208 (44.4)	
**Place of residence; *n* (%)**				
**Vis**	123 (11.2)	110 (8.9)	157 (33.5)	<0.001 †
**Korčula**	727 (66.5)	910 (73.6)	271 (57.9)	
**Split**	244 (22.3)	216 (17.5)	40 (8.5)	
**Education level; *n* (%)**				
**Primary school**	180 (16.6)	294 (24.1)	175 (37.8)	<0.001 †
**Secondary school**	588 (54.1)	633 (51.9)	210 (45.4)	
**University level**	318 (29.3)	293 (24.0)	78 (16.8)	
**Anthropometry**				
**Weight (kg); median (IQR)**	75.9 (21.0)	78.0 (21.2)	79.0 (18.5)	0.001 *
**BMI (kg/m^2^); median (IQR)**	25.5 (5.6)	26.1 (5.6)	26.2 (5.7)	<0.001 *
**WHR; median (IQR)**	0.89 (0.11)	0.92 (0.11)	0.94 (0.11)	<0.001 *
**WHtR; median (IQR)**	0.54 (0.09)	0.55 (0.09)	0.57 (0.10)	<0.001 *
**Metabolic syndrome components**				
**Waist circumference (mm); median (IQR)**	919.5 (158.0)	942.5 (150.0)	967.5 (132.3)	<0.001 *
**Fasting glucose (mmol/L); median (IQR)**	5.2 (0.9)	5.4 (1.0)	5.5 (1.2)	<0.001 *
**Systolic blood pressure; median (IQR)**	125.0 (25.0)	130.0 (20.0)	135.0 (22.6)	<0.001 *
**Diastolic blood pressure; median (IQR)**	80.0 (15.0)	80.0 (10.0)	80.0 (11.6)	<0.001 *
**Triglycerides (mmol/L); median (IQR)**	1.1 (0.7)	1.2 (0.8)	1.3 (0.9)	<0.001 *
**HDL (mmol/L); median (IQR)**	1.5 (0.5)	1.4 (0.5)	1.4 (0.5)	0.351 *
**Other biochemical parameters**				
**Total cholesterol (mmol/L); median (IQR)**	5.8 (1.6)	5.8 (1.6)	5.9 (1.5)	0.284 *
**LDL cholesterol (mmol/L); median (IQR)**	3.7 (1.5)	3.7 (1.3)	3.7 (1.5)	0.911 *
**HbA1c (%); median (IQR)**	5.3 (0.6)	5.3 (0.5)	5.3 (0.6)	0.068 *
**Habits**				
**Smoking; *n* (%)**				
**Never-smokers**	519 (47.7)	602 (49.2)	225 (48.6)	0.833 †
**Ex-smokers**	267 (24.5)	275 (22.5)	111 (24.0)	
**Active smokers**	302 (27.8)	346 (28.3)	127 (27.4)	
**Pack-years in smokers; median (IQR)**	12.0 (18.0)	14.0 (24.0)	20.0 (30.0)	<0.001 *
**Alcohol intake (units/week); median (IQR)**	6.8 (18.9)	6.7 (20.3)	6.8 (20.3)	0.087 *
**Physical activity; *n* (%)**				
**Low**	262 (24.3)	263 (21.6)	103 (22.6)	0.528 †
**Moderate**	717 (66.5)	834 (68.6)	303 (66.4)	
**Intensive**	100 (9.3)	119 (9.8)	50 (11.0)	
**Mediterranean diet adherence (MDSS points); median (IQR)**	11.0 (6.0)	10.0 (5.0)	11.0 (6.0)	0.011 *
**Adding salt before tasting food; *n* (%)**				
**Never**	430 (42.4)	504 (43.6)	139 (31.0)	<0.001 †
**Occasionally**	250 (24.6)	316 (27.3)	102 (22.7)	
**Often**	256 (25.2)	251 (21.7)	164 (36.5)	
**Almost always**	79 (7.8)	86 (7.4)	44 (9.8)	

IQR—interquartile range; BMI—body mass index; WHR—waist-to-hip ratio; WHtR—waist-to-height ratio; MDSS—Mediterranean Diet Serving Score; * Kruskal-Wallis test; † chi-square test.

**Table 2 nutrients-12-01164-t002:** Correlation between age, salt adding habit, threshold and suprathreshold salt taste perception, data presented are Spearman’s rho correlation coefficients (*P* values).

	Salt Taste Threshold	Salt Taste Intensity Perception	Salt Taste Hedonic Perception	Adding Salt Before Food Tasting
**Age**	0.224 (<0.001)	−0.170 (<0.001)	0.038 (0.198)	−0.016 (0.387)
**Salt taste threshold**		−0.132 (<0.001)	0.043 (0.143)	0.092 (<0.001)
**Salt taste intensity perception**			−0.399 (<0.001)	−0.024 (0.424)
**Salt taste hedonic perception**				0.066 (0.025)

**Table 3 nutrients-12-01164-t003:** Prevalence of metabolic syndrome components according to the salt taste recognition threshold perception, data are presented as *n* (%).

	Lower Salt Taste Threshold/Higher Acuity *n* = 1094	Intermediate Salt Taste Threshold/Acuity *n* = 1236	Higher Salt Taste Threshold/Lower Acuity *n* = 468	*p* ^†^
**Elevated waist circumference**	797 (73.5)	980 (79.5)	367 (79.4)	0.001
**Elevated glucose or diabetes present**	90 (8.4)	131 (10.8)	80 (17.4)	<0.001
**Elevated blood pressure or hypertension present**	423 (38.7)	580 (46.9)	263 (56.2)	<0.001
**Elevated triglycerides**	250 (22.9)	340 (27.5)	143 (30.6)	0.002
**Reduced HDL**	207 (18.9)	239 (19.3)	86 (18.4)	0.899
**Metabolic syndrome present**	419 (38.3)	564 (45.6)	271 (57.9)	<0.001

^†^ chi-square test.

**Table 4 nutrients-12-01164-t004:** Association between Mediterranean diet compliance and salt taste threshold and suprathreshold perception.

	Fruit OR (95% CI); *p*	Vegetables OR (95% CI); *p*	Olive Oil OR (95% CI); *p*	Legumes OR (95% CI); *p*	Fish OR (95% CI); *p*	White Meat OR (95% CI); *p*	Red Meat OR (95% CI); *p*	Sweets OR (95% CI); *p*	MDSS Compliance OR (95% CI); *p*
**Salt taste threshold**									
**Higher salt taste threshold (lower acuity)**	referent	referent	referent	referent	referent	referent	referent	referent	referent
**Intermediate salt taste threshold**	1.41 (1.09–1.81); 0.008	0.96 (0.74–1.25); 0.771	0.77 (0.58–1.00); 0.051	0.83 (0.64–1.09); 0.187	0.83 (0.62–1.10); 0.185	0.82 (0.64–1.05); 0.116	0.94 (0.71–1.23); 0.639	1.11 (0.84–1.45); 0.477	0.93 (0.69–1.24); 0.603
**Lower salt taste threshold (higher acuity)**	1.52 (1.16–1.97); 0.002	1.02 (0.78–1.33); 0.881	0.69 (0.52–0.90); 0.008	0.96 (0.72–1.26); 0.749	0.76 (0.58–1.00); 0.053	0.77 (0.61–0.98); 0.035	1.09 (0.82–1.45); 0.544	1.17 (0.88–1.56); 0.295	1.04 (0.77–1.41); 0.807
**Salt taste intensity perception ^§^**									
**No sensation to medium strong**	referent	referent	referent	referent	referent	referent	referent	referent	referent
**Strong to very strong**	0.69 (0.43–1.11); 0.124	1.07 (0.63–1.80); 0.812	1.05 (0.65–1.70); 0.836	0.90 (0.51–1.58); 0.700	1.14 (0.71–1.83); 0.588	1.08 (0.68–1.72); 0.753	1.03 (0.61–1.75); 0.913	0.83 (0.51–1.36); 0.464	0.65 (0.37–1.15); 0.141
**Extremely strong**	1.08 (0.64–1.84); 0.772	1.09 (0.61–1.93); 0.776	1.23 (0.72–2.08); 0.448	1.01 (0.54–1.89); 0.985	1.25 (0.74–2.11); 0.412	1.31 (0.79–2.19); 0.299	1.24 (0.69–2.22); 0.473	0.67 (0.38–1.18); 0.164	0.80 (0.43–1.50); 0.483
**Salt taste hedonic perception ^§^**									
**Dislike**	referent	referent	referent	referent	referent	referent	referent	referent	referent
**Nor like nor dislike**	1.20 (0.84–1.70); 0.320	1.03 (0.71–1.50); 0.884	0.73 (0.52–1.04); 0.081	1.32 (0.88–1.99); 0.175	1.16 (0.81–1.65); 0.430	1.14 (0.81–1.60); 0.452	1.18 (0.81–1.74); 0.383	1.14 (0.79–1.65); 0.470	0.77 (0.48–1.22); 0.263
**Like**	1.02 (0.61–1.71); 0.940	0.84 (0.48–1.48); 0.884	1.65 (0.95–2.88); 0.075	0.91 (0.48–1.71); 0.772	0.89 (0.54–1.48); 0.660	1.23 (0.76–2.01); 0.401	1.19 (0.68–2.08); 0.540	0.77 (0.44–1.37); 0.375	1.21 (0.66–2.21); 0.537

All models were adjusted for age, gender, place of residence, education level (three categories: primary, secondary, university), material status (quartiles), BMI, smoking (three categories: never-smokers, ex-smokers, active smokers), alcohol intake (units/week), physical activity (three categories: low, moderate, intensive), adding salt before tasting food (four categories: never, occasionally, often, almost always); ^§^ calculated from the sample subset including 1155 subjects.

**Table 5 nutrients-12-01164-t005:** Association between metabolic syndrome and its components and salt taste threshold and suprathreshold perception.

	Elevated Waist Circumference OR (95% CI); *p*	Elevated Blood Pressure or Diagnosis of Hypertension OR (95% CI); *p*	Elevated Fasting Plasma Glucose or Diagnosis of Diabetes OR (95% CI); *p*	Elevated Triglycerides or Using Medications OR (95% CI); *p*	Reduced HDL Cholesterol or Using Medications OR (95% CI); *p*	Metabolic Syndrome OR (95% CI); *p*
**Salt taste threshold**						
**Higher salt taste threshold (lower acuity)**	referent	referent	referent	referent	referent	referent
**Intermediate salt taste threshold**	1.03 (0.72–1.46); 0.880	0.85 (0.65–1.13); 0.276	0.71 (0.48–1.06); 0.097	0.89 (0.68–1.19); 0.453	0.65 (0.47–0.91); 0.011	0.75 (0.57–0.99); 0.044
**Lower salt taste threshold (higher acuity)**	0.47 (0.27–0.82); 0.008	0.89 (0.66–1.20); 0.433	0.65 (0.45–0.94); 0.022	0.76 (0.57–1.02); 0.070	0.59 (0.42–0.84); 0.003	0.69 (0.52–0.92); 0.013
**Salt taste intensity perception ^§^**						
**No sensation to medium strong**	referent	referent	referent	referent	referent	referent
**Strong to very strong**	1.17 (0.64–2.17); 0.608	1.57 (0.86–2.77); 0.123	0.97 (0.45–2.11); 0.941	0.89 (0.52–1.52); 0.660	1.11 (059–2.08); 0.743	1.20 (0.81–1.77); 0.362
**Extremely strong**	0.91 (0.47–1.75); 0.766	1.18 (0.63–2.23); 0.602	0.90 (0.36–2.21); 0.812	0.87 (0.48–1.59); 0.648	1.23 (0.63–2.41); 0.550	0.65 (0.35–1.22); 0.177
**Salt taste hedonic perception ^§^**						
**Dislike**	referent	referent	referent	referent	referent	referent
**Neither like nor dislike**	1.04 (0.66–1.63); 0.883	1.13 (0.75–1.70); 0.554	0.74 (0.39–1.41); 0.360	0.78 (0.50–1.20); 0.254	1.01 (0.65–1.56); 0.976	1.35 (0.91–2.01); 0.139
**Like**	0.64 (0.35–1.19); 0.159	1.79 (0.97–3.31); 0.063	0.47 (0.16–1.33); 0.154	1.45 (0.82–2.59); 0.197	0.48 (0.22–1.03); 0.058	1.85 (1.02–3.35); 0.042

All models were adjusted for age, gender, place of residence, education level (three categories: primary, secondary, university), material status (quartiles), BMI (in all regression models except for elevated waist circumference), smoking (three categories: never-smokers, ex-smokers, active smokers), alcohol intake (units/week), physical activity (three categories: low, moderate, intensive), Mediterranean Diet Serving Score (MDSS), adding salt before tasting food (four categories: never, occasionally, often, almost always); ^§^ calculated from the sample subset including 1155 subjects.

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
