# Peer review of "The Association between Salt Taste Perception, Mediterranean Diet and Metabolic Syndrome: A Cross-Sectional Study"

_nutrients, 2020, doi:10.3390/nu12041164_

Round 1
Reviewer 1 Report
The article titled “The Association between Salt Taste Perception, Mediterranean Diet and Metabolic Syndrome: A Cross-Sectional Study” describes relevant results obtained on salt taste perception and Metabolic Syndrome. Specifically, the Authors show that subjects with higher salt taste sensitivity have lower odds of Metabolic Syndrome, for an increased waist circumference, a higher fasting glucose or diabetes, and reduced HDL cholesterol, as compared to the higher threshold group.
The paper is well written. However, there are some points that need to be carefully elucidated. Specific recommendations and questions are detailed below (with section and manuscript lines referenced before each comment).
In section 2 Materials and Methods:
- At page 3, lines 130-131, the Authors state “Material status was based on the subjects’ material possessions and classified into quartiles, as described previously [12]”. I would recommend adding some brief detail.
- At page 4, in section 2.4 “Measurement of salt taste perception” where the Authors describe the taste recognition threshold, they indicate the concentrations used but they do not indicate the method adopted. Did the Authors use the 3-Alternative Forced Choice (3-AFC) or other? Please clarify.
- At page 5, line 200, the Authors indicate the concentration used for suprathreshold salt taste perception. How did the Author choose this concentration?
In section 3 Results
- I would recommend adding a table with subjects’ characteristics, as in table 1, dividing the subjects who have Metabolic Syndrome and subjects with no Metabolic Syndrome.
- Did the Authors perform the analysis considering only the subjects with Metabolic Syndrome? Otherwise, I would recommend considering this point and perform the same analysis used for all subjects, but including only the MetS subjects (44%).
- In table 1, are the values of diastolic blood pressure correct? The Authors indicate 80 as the mean value for all of three groups and the p value is <0.0001. How is it possible? Please verify.
- In table 5, the Authors present the association between Metabolic Syndrome and its components and salt taste perception. This table could be shortened, focusing on data that show statistical significance (as the case of Model 2), while data that did not achieve significance should be referred to in the main body of the text. This would allow the reader to easily identify the most important data which support the overall conclusions of this investigation.
In section 4 Discussion
- At page 17, lines 373-377, the Authors discuss the results of salt taste threshold and MetS. Where do the percentages come from? Are the Authors explaining the results shown in table 3? The Authors state: “.. subjects with lower salt taste threshold, indicating higher salt taste sensitivity, had 31% lower odds of having MetS…”: if this result refers to table 3 the correct value should be 38.3%. Please verify and clarify.
- Please reduce the discussion to no more than 3 pages: it is too long, and some parts are not strictly necessary. For example, in lines 362-365, 415-422 and 442-455, the Authors included results on sweet, bitter and umami which are not relevant in this paper. I would suggest deleting these parts. I would recommend focusing on what new insights your results provide.
Author Response
Point 1: At page 3, lines 130-131, the Authors state “Material status was based on the subjects’ material possessions and classified into quartiles, as described previously [12]”. I would recommend adding some brief detail.
Response 1: We have added this description (page 3, lines 140-144):
„Namely, subjects answered to 16 questions about their material possessions (heating system, wooden floors, video/DVD recorder, telephone, computer, two TVs, freezer, dishwasher, water supply system, flushing toilet, bathroom, library with more than 100 books, paintings or other art objects, a car, vacation house or second apartment, boat). Based on the sum of all positive responses, subjects were divided into quartiles.“
Point 2: At page 4, in section 2.4 “Measurement of salt taste perception” where the Authors describe the taste recognition threshold, they indicate the concentrations used but they do not indicate the method adopted. Did the Authors use the 3-Alternative Forced Choice (3-AFC) or other? Please clarify.
Response 2: We added this description (page 5, lines 204-209):
„Subjects were presented with only one solution at a time in order to make the testing procedure overall less time consuming and less cumbersome, especially for elderly subjects. With each solution presentation, subjects were asked whether the solution tasted like plane water or something else, and in case of confirmatory answer, they were asked to identify the quality of the taste (sweet, salty, sour or bitter). Because of this simplified testing procedure, unlike the usual 3-alternative forced choice or 2-alternative forced choice, each subject performed two recognition threshold testing rounds, with a break of at least of 30 minutes in between them.“
Point 3: At page 5, line 200, the Authors indicate the concentration used for suprathreshold salt taste perception. How did the Author choose this concentration?
Response 3: We added this sentence as an explanation (lines 219-222):
“This concentration is slightly more than double the highest threshold solution concentration (3.7 g/L), and it was used before as the highest concentration for threshold testing [37], pointing to be a possible concentration for suprathreshold testing, but not too concentrated to be off-putting”.
Point 4: I would recommend adding a table with subjects’ characteristics, as in table 1, dividing the subjects who have Metabolic Syndrome and subjects with no Metabolic Syndrome
Response 4: We have added the requested analysis as an additional Supplementary Table 1, alongside with the description of the table (lines 293-303).
Point 5: Did the Authors perform the analysis considering only the subjects with Metabolic Syndrome? Otherwise, I would recommend considering this point and perform the same analysis used for all subjects, but including only the MetS subjects (44%).
Response 5: We have included both subjects with metabolic syndrome and those without it, where the group without metabolic syndrome served as the control group. If we would completely exclude subjects without metabolic syndrome from the analysis, we would not be able to answer the question whether there is an association between salt taste perception and the presence of metabolic syndrome. However, we have now present characteristics of subjects with metabolic syndrome and those without it in a new Supplementary Table 1. We hope that adding all the relevant information in this table, we have answered to this point raised by the reviewer.
Point 6: In table 1, are the values of diastolic blood pressure correct? The Authors indicate 80 as the mean value for all of three groups and the p-value is <0.0001. How is it possible? Please verify.
Response 6: This is due to the non-normal data distribution and even though the medians are the same, the distribution is skewed more to the right for subjects with intermediate and higher salt taste threshold.
Point 7: In table 5, the Authors present the association between Metabolic Syndrome and its components and salt taste perception. This table could be shortened, focusing on data that show statistical significance (as the case of Model 2), while data that did not achieve significance should be referred to in the main body of the text. This would allow the reader to easily identify the most important data which support the overall conclusions of this investigation.
Response 7: We have carefully considered this comment, and we have deleted the results obtained in Model 1 from this table, and left only results obtained in fully adjusted Model 2. Because of this, we have also changed the Methods section (we deleted the description of Model 1) and in the Results section.
Point 8: At page 17, lines 373-377, the Authors discuss the results of salt taste threshold and MetS. Where do the percentages come from? Are the Authors explaining the results shown in table 3? The Authors state: “.. subjects with lower salt taste threshold, indicating higher salt taste sensitivity, had 31% lower odds of having MetS…”: if this result refers to table 3 the correct value should be 38.3%. Please verify and clarify.
Response 8: These numbers come from regression analysis results, presented in Table 5. Since OR was 0.69 for metabolic syndrome in subjects with lower salt taste threshold compared to higher salt taste threshold, and it was a significant result (P=0.013), we subtracted 0.69 from 1 (no difference), to get a 31% reduction in odds for having metabolic syndrome. To make it clearly separate from results presented in Table 3, we have added this info (lines 418-424 ):
“Namely, based on the regression analysis, we found that subjects with lower salt taste threshold, indicating higher salt taste sensitivity, had 31% lower odds of having MetS (OR=0.69; 95% CI 0.52-0.92). They also had 53% lower odds for elevated waist circumference (OR=0.47; 95% CI 0.27-0.82), 35% lower odds for having elevated fasting glucose or diabetes diagnosis (OR=0.65; 95% CI 0.45-0.94), and 41% lower odds for having reduced HDL cholesterol (OR=0.59; 95% CI 0.42-0.84), compared to the higher threshold group, while the result for elevated triglycerides was borderline insignificant (OR=0.76; 95% CI 0.57-1.02).”
Point 9: Please reduce the discussion to no more than 3 pages: it is too long, and some parts are not strictly necessary. For example, in lines 362-365, 415-422 and 442-455, the Authors included results on sweet, bitter and umami which are not relevant in this paper. I would suggest deleting these parts. I would recommend focusing on what new insights your results provide.-
Response 9: We have deleted these paragraphs in order to shorten the discussion.
Reviewer 2 Report
Although this reviewer warmly welcomes this manuscript, some questions should be addressed:
-The rationale for the study is unclear as the introduction is a bit confusing, comprising several pieces of apparently unlinked information. A more integrated appraisal of the relevant literature would be appropriate to provide the context for the study.
-The importance of food processing and supply chain length in the pathogenesis of MetS (Coscioni et al. J. Clin. Med. 2019, 8(12), 2061; PMID: 31771147) should be better addressed.
-It is advisable to the Authors to incorporate a pictorial or cartoon representation of the main results of the study to increase the overall impact of the manuscript.
-The strengths and limitations of the study should be deeply addressed, taking into account sources of potential bias or imprecision: Discuss both direction and magnitude of any potential bias.
Author Response
Point 1: The rationale for the study is unclear as the introduction is a bit confusing, comprising several pieces of apparently unlinked information. A more integrated appraisal of the relevant literature would be appropriate to provide the context for the study.
Response 1: We have shorten the first part of the introduction and we tried to create a better “flow” of the text.
Point 2: The importance of food processing and supply chain length in the pathogenesis of MetS (Coscioni et al. J. Clin. Med. 2019, 8(12), 2061; PMID: 31771147) should be better addressed.
Response 2: We have added this reference in the context of the Mediterranean diet and its association with the metabolic syndrome (lines 77-82). We noticed that due to track changes, we could not update our EndNote reference library properly, that is why, we have added the full reference to the introduction text, so that the track changes we made remain visible.
Point 3: It is advisable to the Authors to incorporate a pictorial or cartoon representation of the main results of the study to increase the overall impact of the manuscript.
Response 3: We are not sure what is meant by “pictorial or cartoon representation”. Still, we provide a figure, which summarizes our main results (lines 381- 383).
Point 4: The strengths and limitations of the study should be deeply addressed, taking into account sources of potential bias or imprecision: Discuss both direction and magnitude of any potential bias.
Response 4: We have stated main limitations of the study. Based on this comment, we have expanded this part of the discussion (lines 566-584):
“Several limitations and advantages of this study should be considered. Firstly, this is a cross-sectional study, and therefore causality cannot be assessed in determining taste differences as a cause of MetS. Namely, we cannot elucidate whether subjects with reduced salt taste acuity (increased threshold) developed MetS as a consequence of this sensory characteristic or the MetS was is a causal predictor or a consequence of the salt taste acuity loss MetS. For example, subjects with lower salt taste acuity could also have altered perception of food, which could influence their food choices toward less healthy, more processed salty foods, influencing their body weight and the risk for MetS development. Unfortunately, these habits were not included in the questionnaire, and we can not clarify this further based on the available data. Secondly, data for salt taste intensity and hedonic perception were available only for a subset of subjects, which might have introduced bias of undetermined direction and magnitude. The conditions of the field-testing were not ideal as would have been in the laboratory, which could have resulted in less accurate and precise measures of salt taste perception. Lastly, we used aqueous solution to deliver testant, while most of the salty foods are not consumed in a liquid form, which might have influenced suprathreshold responses and especially hedonic responses of the subjects. For instance, subjects could have rated concentrated salty solution as more intense and less hedonically appealing, compared to the rating they would provide for solid salty foods, such as chips or cured processed meats. This could be behind our result of absent association between suprathreshold salt taste perception and prevalent MetS and its components.”
Reviewer 3 Report
Dear authors,
Thank you for your interesting work presented to this journal entitled The Association between Salt Taste Perception, Mediterranean Diet and Metabolic Syndrome: A Cross-Sectional Study. Taste loss could be a metabolic consequence of the obese state and also greater sodium consumption could contribute to higher body weight. The objective of this paper was to examine the association between salt taste threshold and suprathreshold perception and MetS components in the general population of Dalmatia, Croatia. Additionally, te authors examined the adherence to the Mediterranean diet according to the salt taste perception. This is the first study to examine the association between salt taste threshold sensitivity, intensity perception and hedonic rating, with both MetS and the Mediterranean diet. Salt taste intensity perception did not show significant association with metabolic syndrome. The study showed that subjects with higher salt taste threshold were on average older than those with lower threshold. there was no difference in the overall compliance to the Mediterranean diet between lower and higher salt taste threshold group.
The study is well designed and well conducted, congratulations. There is only a little concern, in lines 393, 429 and 453 is necessary include a dot in de references: et al.
Author Response
Point 1: The study is well designed and well conducted, congratulations. There is only a little concern, in lines 393, 429 and 453 is necessary include a dot in de references: et al.
Response 1: Thank you, we have corrected this.
Round 2
Reviewer 1 Report
I appreciate the changes that the authors have made following my suggestions. The manuscript is now suitable for publication.